# Modulation of Ferroptosis by microRNAs in Human Cancer

**DOI:** 10.3390/jpm13050719

**Published:** 2023-04-24

**Authors:** Irena Velkova, Martina Pasino, Zumama Khalid, Paola Menichini, Emanuele Martorana, Alberto Izzotti, Alessandra Pulliero

**Affiliations:** 1IRCCS Ospedale Policlinico San Martino, 16132 Genova, Italy; irena_velkova@hotmail.it (I.V.); martina.pasino01@gmail.com (M.P.);; 2Department of Health Sciences, University of Genoa, 16132 Genoa, Italy; zumama.khalid@edu.unige.it; 3Istituto Oncologico del Mediterraneo, 95029 Viagrande, Italy; 4Department of Experimental Medicine, University of Genoa, 16132 Genoa, Italy

**Keywords:** ferroptosis, microRNA, human cancer, ROS

## Abstract

Ferroptosis is a cell death pathway triggered by an imbalance between the production of oxidants and antioxidants, which plays an emerging role in tumorigenesis. It is mainly regulated at three different levels including iron metabolism, the antioxidant response, and lipid metabolism. Epigenetic dysregulation is a “hallmark” of human cancer, with nearly half of all human cancers harboring mutations in epigenetic regulators such as microRNA. While being the crucial player in controlling gene expression at the mRNA level, microRNAs have recently been shown to modulate cancer growth and development via the ferroptosis pathway. In this scenario, some miRNAs have a function in upregulating, while others play a role in inhibiting ferroptosis activity. The investigation of validated targets using the miRBase, miRTarBase, and miRecords platforms identified 13 genes that appeared enriched for iron metabolism, lipid peroxidation, and antioxidant defense; all are recognized contributors of tumoral suppression or progression phenotypes. This review summarizes and discuss the mechanism by which ferroptosis is initiated through an imbalance in the three pathways, the potential function of microRNAs in the control of this process, and a description of the treatments that have been shown to have an impact on the ferroptosis in cancer along with potential novel effects.

## 1. Introduction

MicroRNAs (miRNAs), defined as small noncoding RNAs of around 25 nucleotides that regulate gene expression at the post-transcriptional level, were initially described in 2005; currently, over 2700 mature miRNAs have been identified in humans with characteristic expression profiles [1]. miRNAs are often correlated with pathologic conditions and are involved in the regulation of biological processes and normal cell functions. Indeed, targeting miRNA was explored as a therapeutic option [2]. In cancer, many miRNAs are associated with apoptosis evasion, cell proliferation, angiogenesis, metastasis, and more recently ferroptosis; therefore, these molecules are classified according to the function of their mRNA targets [3]. The modulation of miRNAs’ expression levels can activate oncogenes (oncomiRs) or inactivate tumor suppressor genes, contributing to the initiation and progression of cancer. The production of miRNAs is a strictly transcriptional process. The nuclear-coding gene, which is mostly located in intron regions, is transcribed into primary transcripts (pri-miRNA) by binding to RNA Polymerase III. Drosha and Dicer, as two types of Ribonuclease III, play an important role in the pri-miRNA maturation. Exportin-5 is a Ras-related nuclear protein-Guanosine Triphosphate complex. Pre-miRNA is transported by it from the nucleus to the cytoplasm. With the help of Trans-Activation Response RNA-Binding Protein, Dicer splices the pre-miRNA into a miRNA duplex in the cytoplasm [3].

Many diseases are associated with ferroptosis. Inhibiting or promoting the enzymes or genes involved in ferroptosis has an impact on the course of diseases. miRNAs can alter disease progression by modulating ferroptosis. Interestingly, the role of ferroptosis in cancer is not quite the same as in other diseases. Ferroptosis can inhibit the proliferation and cell cycle of cancer cells, which suggests that we should focus on ferroptosis-based therapies for cancer [4].

Ferroptosis is caused by a redox imbalance between the production of oxidants and antioxidants, which is driven by the different expressions and activities of the multiple redox-active enzymes that produce or detoxify free radicals and lipid oxidation products. Accordingly, ferroptosis is regulated at multiple levels, including epigenetic, transcriptional, post-transcriptional, and post-translational levels [5]. Different studies suggest that ferroptosis plays a pivotal role in tumor suppression, thus providing new opportunities for cancer therapy.

The present review discusses the microRNA modulation related to ferroptosis with the aim of providing novel insight to increase knowledge about personalized cancer therapy.

## 2. Ferroptotic Pathway

Ferroptosis is an iron-dependent form of programmed cell death. Defined by a novel term for the first time by Dixon et al. in 2012 [6], it has been studied before [7,8,9] and has been described in the last decade as a different type of cell death with respect to apoptosis, necrosis, and autophagy, from which it differs from genetic, biochemical, and morphological point of views.

The term “ferroptosis” was coined following the observations that RAS-mutated cancer cells were more sensitive to ferroptosis activation compared to cancer cells that lack RAS mutations [6,10]. This could be explained by the tight connection between oncogenic RAS activation and the high levels of intracellular iron in these cells, since the oncogenic RAS modulates iron metabolism through the regulation of transferrin receptor 1 (TFR1) [9,11,12]. However, it has been observed that, while mutated RAS could be necessary to initiate the ferroptosis mechanism, several kinds of cancer are also sensitive to ferroptosis induction, even if they lack RAS mutations [13]. In contrast, two important inducers of ferroptosis such as Erastin and RSL3 (oncogenic-RAS-selective lethal compounds) are known to specifically induce this kind of cell death in RAS-mutant cancer cells following the accumulation of reactive oxygen species by an iron-dependent mechanism [14,15].

Recent studies reported that the well-known tumor suppressor gene p53 plays a role in ferroptosis [16]. P53 was first reported to sensitize cells to ferroptosis through transcriptionally repressing SLC7A11 which representsone of its direct targets [17]. The regulation of ferroptisis by p53 contributes to the tumor suppressive function of p53 itself. Notably, tumors expressing mutant p53 may display decreased levels of SLC7A11 and increased sensitivity to ferroptosis [18]. The role of p53 in ferroptosis is rather complex implying both a promotion as well as a suppression of the process, likely in a cell-context-dependent manner and involving different finely tuned proteins and pathways with regulation that is not discussed in this review.

Ferroptosis is the consequence of an imbalance between the levels of reactive oxygen species (ROS) and the activity of the antioxidant defense, with a consequent accumulation of ROS in the plasmatic membrane, process widely known for causing many disorders in the cell [19]. In this context, a significant intermediate role is played by the different components of iron metabolism. Therefore, there are three mechanisms that, when deregulated, have a key role in triggering the ferroptosis machinery: iron metabolism, the antioxidant defense, and lipid metabolism (Figure 1).

### 2.1. Iron Metabolism

Ferroptosis is an iron-dependent type of cell death; therefore, the intracellular levels of iron play a major role in its induction. The crucial mediators are a group of factors that regulate iron at different levels inside and outside the cell. In the blood, iron circulates in the Fe^3+^ form carried by a special transport protein called transferrin (TF). When it must enter the cell, Fe^3+^ is carried by transferrin near the plasmatic membrane, where the transferrin receptor 1 (TFR1) recognizes the complex and allows the Fe^3+^ to enter by endocytosis. Inside the endosome, Fe^3+^ is reduced to Fe^2+^ by the metal reductase Six-Transmembrane Epithelial Antigen of Prostate 3 (STEAP3) [20]. Fe^2+^ can then be released in the cytoplasm by the Divalent Metal Transporter (DMT1) and enter the labile iron pool (LIP) [21]. When the Fe^2+^ in the LIP reaches high levels, it can be stored mainly in ferritin or as heme.

Ferritin is an iron storage protein complex formed by two subunits consisting of a ferritin light chain (FTL) and a ferritin heavy chain 1 (FTH1) [22,23]. The FTH1 subunit has a ferroxidase activity that is able to oxidase the ferrous iron into the ferric form in which it can be stored [24,25]. The mediator that regulates the release of Fe^2+^ from the ferritin storage is the nuclear receptor co-activator 4 (NCOA4) that is able to bind the heavy chain of the ferritin complex and promote the ferritin degradation mediated by the lysosomes (“ferritinophagy”) [26,27]. Any overactivity of the NCOA4 can cause a major ferritin degradation, leading to an increase in the intracellular LIP. In contrast, some type of cells, such as macrophages, enterocytes, red blood cells, and neurons, export iron on the outside; only one iron exporter is known, called ferroportin [28].

The regulation of intracellular iron levels is controlled by iron-responsive element-binding proteins IRP1 and IRP2 [29]: when the iron decreases, IRP1 and IRP2 can bind the iron-responsive elements (IREs) located on the UTR regions of the mRNA [30]. Ferritin and ferroportin contain the IREs in the UTR regions, so when the cell is in a state of iron deficiency, IRPs repress their synthesis [30].

The other iron storage protein, heme, can release the Fe^2+^ when induced by heme oxygenase-1 (HO-1) [31,32]. The increase in Fe^2+^ in the cellular LIP can be one of the first events triggering the ferroptosis machinery, since ferrous iron can easily be used in the Fenton reaction to produce free radicals such as hydroxyl radicals [33]. In this way, the iron can participate directly in the peroxidation of phospholipids (PLOOH) that subsequently will lead to the activation of the whole ferroptosis process. Furthermore, iron is the mineral that catalyzes many important physiological reactions in the cell, such as the formation of most ROS, which can in turn contribute to lipid peroxidation and, thus, to ferroptosis [34].

### 2.2. Antioxidant Defense

Two years after the first definition of ferroptosis, in 2014, Yang et al. identified GPX4 as a protein playing a crucial role in this mechanism [35], although other GPX4-independent pathways were also described. GPX4 is an antioxidant enzyme that can reduce phospholipid hydroperoxide to corresponding alcohols in the cell membrane; therefore, it is considered the key element of ferroptosis inhibition. GPX4 is a selenoprotein that uses reduced glutathione (GSH) as an essential substrate; thus, it is dependent on GSH availability for its proper functioning [36] and consists of three amino acids, glutamate, cysteine, and glycine, with a binding that is mediated by glutamate-cysteine ligase and glutathione synthetase. The intracellular level of GSH is strongly regulated by the uptake of cysteine, which is internalized as cystine via the so-called Xc-complex, a glutamate-cystine membrane antiport, consisting of the SLC7A11 (xCT) and SLC3A2 (4F2hc) subunits [37,38].

The deprivation of cysteine in the cell causes GSH depletion, which can inhibit the GPX4 function, thus leading to the accumulation of lipid hydroperoxides and to ferroptosis due to membrane damage. On the other hand, GPX4-independent pathways were described such as the FSP1 (ferroptosis inhibitor protein 1)-CoQ antioxidant complex that functions only in GPX4-deprived cells. In the membrane, FSP1 reduces ubiquinone to ubiquinol to limit the increase in ROS in the plasma membrane [39,40].

### 2.3. Lipid Metabolism

The lipid metabolism is the third pathway that comes into play to induce ferroptosis through lipid peroxidation. The final products of this process, such as malondialdehyde (MDA), cause membrane instability through damage of the lipid bilayer, which increases the cell permeability and can lead to cell death [37].

The major players of this pathway are the Polyunsaturated Fatty Acids (PUFA), because they are more prone to go into peroxidation especially during ferroptosis development.

PUFAs can be synthesized starting from the acyl-CoA synthetase long-chain family member 4 (ACSL4) that binds PUFA to coenzyme A (CoA) to produce acyl-CoA. Afterward, thanks to the action of several acyltransferases, acyl-CoA can be re-esterified in phospholipids [41,42]. Linoleic acid and arachidonic acid are the most abundant PUFAs in the cell that can be oxidized into hydroperoxides by lipoxygenase (LOX) action. Indeed, LOXs are enzymes that contain the iron element and use it as a cofactor for catalyzing the deoxygenation of both esterified and not esterified PUFAs to produce lipid hydroperoxide in a pro-ferroptosis full picture [11,35,43]. Thus, iron metabolism (i.e., iron levels), antioxidant actions, and lipid metabolism (i.e., phospholipid peroxidation) each have an essential role in triggering critical changes in the cell, leading to cell death through ferroptosis. Regarding the many levels in which ferroptosis can be regulated, the role of non-coding RNAs and especially microRNAs may represent a crucial determinant to finely tune this cell death pathway.

## 3. Ferroptosis and Cancer Therapy

The complexity of the mechanisms that regulate tumorigenesis and tumor progression leads to the difficult eradication of cancer cells. Apoptosis is one of the principal targets for cancer treatments; however, in the last few years, many observations indicate ferroptosis as a promising pathway to be investigated for cancer therapy. As previously discussed, iron homeostasis, the Xc-system, and lipid metabolism all play a central role in promoting ferroptosis [15]. Therefore, many molecules are being studied to target and modulate these pathways in a tumor-suppressive manner.

### 3.1. Targeting Iron Metabolism

Cancer cells require an increased amount of iron for survival in comparison to normal cells [44]; thus, the modulation of iron metabolism can be exploited to induce cell death by ferroptosis. A variety of compounds could be used for this purpose, each of them effective at different levels of the ferroptosis machinery. Siramesine, a lysosome disrupting agent, and Lapatinib, a tyrosine kinase inhibitor, both induce an increase in the intracellular iron level because they trigger the upregulation of transferrin (TF) and the downregulation of ferroportin-1 [45]. The inhibition of the iron transport system causes an increase in ROS and ferroptosis in breast cancer [46].

Dihydroartemisinin is the semi-synthetic derivative of Artemisinin, first used as an antimalarial drug [47]. Chen et al. observed that this molecule can associate with the cell’s free iron, allowing it to maintain its oxidative activity. By inducing the lysosomal degradation of ferritin, Dihydroartemisinin increases the cellular free iron level, thus sensitizing cells to ferroptosis in glioblastoma, breast cancer, lung cancer, and colorectal cancer cells [48]. Artesunate, another derivative of Artemisinin, accumulates in lysosomes, reacts with iron, and causes ferritin degradation with a mechanism analogous to Dihydroartemisinin. This was exploited for ferroptosis induction in ovarian cancer, pancreatic cancer, and glioblastoma [49].

### 3.2. Targeting the Xc-System and Glutathione Axis

Glutathione synthesis is essential to maintain a reduced state in the cellular environment, avoiding lipid peroxidation and ferroptosis as the final effect. Since SLC7A11, GPX4, and GSH are the focal points of this pathway, an increasing number of compounds are being studied to target these proteins. These are ferroptosis inducers (FINs), a set of molecules categorized into four classes according to their different mechanisms of action [50].

Class I FINs aim to block GSH synthesis by directly inhibiting SLC7A11 and consequently cysteine uptake [51]. This group includes Erastin, its analogue Imidazole Ketone Erastin, Sulfasalazine, and Sorafenib. The latter two are FDA-approved agents with anti-inflammatory and multi-kinase inhibitor effects, respectively, in addition to class I FIN activity. Sulfasalazine’s effect was already demonstrated in patients with glioblastoma, and a phase I clinical trial, to test its combination with radiotherapy, is ongoing (NCT04205357). Erastin is widely used in cell culture; however, its low water solubility and metabolic stability represent limitations for in vivo applications [52]. In addition, studies in mice revealed the induction of some Erastin-induced toxicity such as the alteration of blood index values, causing mild cerebral infarction of the brain and enlarged glomerular volume of the kidney [53]. Reduced systemic toxicity is obtained with the Imidazole Ketone Erastin, allowing its use for in vivo experiments with good results in terms of ferroptosis induction [54]. All these compounds are being studied in clinical trials at different phases, for the treatment of lymphomas, pancreatic cancer, lung cancer, glioblastoma, and hepatocellular and renal cell carcinoma [15,51,53]. GPX4 is highly expressed in cancer cells where it can arrest ferroptosis, enhancing the anticancer effects of cisplatin [55]. Recently, it was observed that cisplatin can also stimulate ferroptosis in lung and colorectal cancer cells: the high affinity to thiol groups leads to its conjugation with the intracellular glutathione, thus reducing the cellular antioxidant defense and enhancing ferroptosis [56]. This finding opens new perspectives to overcome cisplatin resistance by combining this agent with pro-oxidant molecules, such as Erastin [55].

Class II FINs target GPX4 by covalently binding selenocysteine in its catalytic site [37,42]. RSL3, ML162, and ML210 belong to this class but show a low solubility, which makes them unsuitable for in vivo applications [57]. Altretamine (FDA-approved) and Withaferin A (a natural anticancer compound) both inhibit GPX4 activity or reduce its levels, thus appearing to be interesting alternatives for the in vivo treatment of ovarian cancer [58].

Class III FINs are represented by statins. They are hydroxy methyl glutaryl-CoA inhibitors and cause a depletion of GPX4 and CoQ10, but the hypothetical liaison between cholesterol and ferroptosis is still unclear [50]. A novel type III FIN, Fin56, was recently developed. It can induce ferroptosis through the degradation of GPX4, but the mechanisms, which likely involve autophagy, are still not clear [59].

Class IV FINs, such as FINO2, are compounds in a preliminary experimental stage and primarily act by increasing the oxidizing iron and then inactivating GPX4. Both class III and IV FINs have not yet been tested in vivo [52]. The in vivo use of various FINs is limited by their low solubility and unsatisfactory pharmacokinetics. Thus, their effective use in clinical treatment remains an open challenge [50].

It should also be considered that standard therapies (radiotherapy and chemotherapy) could induce ferroptosis [60]. For instance, following radiotherapy, cancer cells may undergo an adaptive response by increasing SLC7A11 and GPX4 levels [61]. In light of this, the use of traditional cancer therapy agents in combination with FINs can enhance the outcome of ferroptosis-inducing treatment [62]. It is worth noting that the combination of these two classes of molecules has minimal toxicity for healthy cells, thus supporting this rationale [63].

### 3.3. Targeting Lipids Metabolism

Lipid peroxidation can be caused by multiple mechanisms, including GSH depletion and GPX4 inhibition [50]. It is evident that the exacerbation of these mechanisms has, as the final effect, the increase in lipid ROS and consequently the induction of ferroptosis. In fact, statins (class III FINs) and FINO2 (class IV FINs) both lead to lipid peroxidation through GPX4 inhibition and iron oxidation [64,65]. To increase the level of lipid ROS, it is necessary to act on upstream pathways; thus, an alternative route not necessarily based on the glutathione system should be considered. In this respect, a pathway that can be targeted to enhance lipid ROS accumulation is the respiratory chain. BAY-87-2243 becomes relevant in this sense; in fact, it inhibits complex I of the mitochondrial respiratory chain, thus stimulating ROS generation, lipid peroxidation, and ferroptosis as the final event [66].

## 4. Role of MicroRNAs in Ferroptosis

### 4.1. MicroRNAs in Ferroptosis Activation

While being the crucial player in controlling gene expression at the mRNA level, microRNAs (miRNAs) were recently shown to modulate cancer growth and development via the ferroptosis pathway [67,68].

Whether by controlling GPX4, FSP1, or GSH, microRNAs fundamentally influence tumor ferroptosis via the antioxidant system. GPX4 is the most significant antioxidant component in the ferroptosis-regulation network. For instance, microRNAs have GPX4 as a primary target in hematological malignancies [69]. Additionally, MiR-15a-3p was discovered to play a role in ferroptosis in colorectal cancer [67]. It can specifically bind to the 3′-UTR of GPX4 and limit its action, which raises cytoplasmic ROS, intracellular Fe^2+^, and MDA levels. Inhibiting GPX4 causes the antioxidant system to be disrupted, making it impossible for the lipid peroxides to be cleared in a timely manner from the cells, which would trigger ferroptosis [68]. Research by Xu et al. shows that miR-15a overexpression can stop prostate cancer cells from proliferating; boost the production of lactate dehydrogenase, MDA, Fe^2+^, and ROS; and subsequently destroy MMP (mitochondrial membrane potential) by blocking GPX4 [70]. A systematic study performed on miR-1287-5p revealed that it directly binds to the 3′ UTR of GPX4 to suppress its protein activity, since the overexpression of GPX4 entirely stops miR-1287-5p-induced ferroptosis and tumor suppression [71]. The overexpression of miR-1287-5p significantly induces ferroptosis, whereas its inhibition suppresses ferroptosis in osteosarcoma cells. GPX4 is also considered a direct target of miR-324-3p, and the overexpression of GPX4 reverses its effect. MicroRNAs act by modulating many different biological processes, including chemoresistance. In A549 cells, miR-324-3p enhances cisplatin-induced ferroptosis [72]. A study performed by Sun et al. showed that the overexpression of miR-34c-3p in squamous cell carcinoma cells inhibits cell proliferation and in turn promotes ferroptosis by increasing ROS, MDA, and iron and reducing GSH and GPX4 levels [73].

Another crucial regulatory component of the ferroptosis-signaling cascade is SLC7A11. Recently, it was discovered that several miRNAs target SLC7A11 to control the ferroptosis process in tumor cells through antioxidant pathways and by targeting 3′UTR [74]. It is hypothesized that miR-5096 can target and downregulate SLC7A11 through an in silico method. SLC7A11 is a target of miR-5096, as demonstrated by a 3’UTR luciferase assay, and its target position was further confirmed by the discovery of decreased SLC7A11 miRNA and protein levels in response to miR-5096 overexpression. By simultaneously increasing ROS, OH-, lipid ROS, and iron accumulation levels and decreasing GSH and mitochondrial membrane potential with mitochondrial downsizing and significant cristae loss, miR-5096 induces ferroptosis cell death in breast cancer [74]. MiR-378a-3p was seen regulating SLC7A11 and eventually promoting ferroptosis through a MiR-378a-3p/SLC7A11 axis in a nerve injury [75]. In gastric cancer as well, miR-375 was seen targeting SLC7A11, thus inducing ferroptosis [69,76]. A study performed on a renal cell carcinoma showed that the silencing of SLC7A11 induces ferroptosis through miR-143-3p signaling [77]. In a study with in vivo and in vitro models of Parkinson’s disease, it was seen that miR-335 promotes ferroptosis by targeting ferritin heavy chain 1 (FTH1) [78]. Through the upregulation of IREB2, oxidative stress and ferroptosis were found to increase with the decreased expression of miR-122 [79].

### 4.2. MicroRNAs in Ferroptosis Inhibition

Ferroptosis is directly linked to long-chain acyl-CoA synthetase 4 (ACSL4). In hepatocellular carcinoma (HCC), miR-23a-3b is considered as a prominent miRNA, and ACSL4 is found to be a target gene in Sorafenib-resistant cells in HCC. MiR-23a-3p targets the 3′-UTR of ACSL4 and inhibits ferroptosis [80]. Ferroptosis is implicated in the initiation and progression of various tumors, including glioblastoma. MiR-670-3p was found to suppress ferroptosis through lipid metabolism, targeting the ACSL4 gene in glioblastoma. An inhibitor of this miRNA has a reverse effect; thus, the induction of ferroptosis by inhibiting miR-670-3p can be considered an adjuvant strategy to treat glioblastoma [81]. Similarly, miR-424-5p suppresses ferroptosis by targeting ACSL4 gene in ovarian cancer. The decrease in lipid peroxidation and ferroptosis mediated by miR-424-5p can be abolished by ACSL4 overexpression, and vice versa [82,83].

A study performed by Liao et al. showed miR-130b-3p to be having an inhibitory function in Erastin- and RSL3-induced ferroptosis, as demonstrated by a decrease in lipid peroxidation and ferrous ion content. miR-130b-3p triggered the Nrf2/HO-1 pathway by reducing the expression of its target gene, DKK1, hence regulating ferroptosis. The ability of Erastin to fight tumors was inhibited by miR-130b-3p. An in vivo mice model was used to replicate in vitro findings; it was found that miR-130b-3p inhibition decreased the tumor volume, while miR-130b-3p ectopic expression increased the tumor volume in those mice models. In mice G-361 cells, Erastin’s anticancer activity was consistently reduced by miR-130b-3p. These findings point to miR-130b-3p’s capacity to suppress ferroptosis in melanoma cells, through the Nrf2/HO-1 pathway [84,85].

Ferroptosis-related gene SLC1A5 is a novel prognostic marker linked to ferroptosis cell death, although it was also studied as a negative regulator of ferroptosis in some cases. The overexpression of miR-137 prevents ferroptosis, suppresses iron build-up, and reduces lipid peroxidation by inhibiting the SLC1A5 glutamine transporter expression. However, the overexpression of SLC1A5 counteracts miR-137’s protective action against ferroptosis. By focusing on the control of the glutamine transporter SLC1A5 as a novel therapeutic target for melanoma, miR-137 can inhibit ferroptosis in melanoma cells [86]. Another miRNA, namely miR-19a, was described as a ferroptosis suppressor through the inhibition of IREB2 [87].

Lipoxygenases (LOXs) are a family of enzymes that produce oxylipin from polyunsaturated fatty acids. The LOXs consist of six isoforms including ALOXE3. In glioblastoma cells (GBM), ALOXE3 was identified as a tumor promoter. miR-18 was observed as a ferroptosis suppressor, which regulates ALOXE3 through the YAP signaling pathway. MiR-18 downregulates ALOXE3 and increases glioblastoma development and migration activity [88,89].

The GLS2 gene was studied as both a positive and negative regulator of ferroptosis in different cases. In one study, miR-190a-5p was seen as a negative regulator of ferroptosis, which directly targets the GLS2 gene. In cardiomyocytes, the overexpression of miR-190a-5p was studied as an inhibiting factor for GLS2, which eventually downregulates ROS, Fe^2+^, and MDA [90]. It was shown by Zheng et al. that the inhibition of miR-545 reduces ferroptosis in colorectal cancer (CRC) by targeting Transferrin (TF). The overexpression of TF was observed as blocking miR-545 by inducing changes in ROS, Fe^2+^, and MDA in the CRC cells, thus stimulating CRC cell death and suppressing ferroptosis [91].

CircKIF4Ab was found to be an enhancer of the GPX4 being targeted by miR-1231 [92]. Contrarily, miR-182-5p, miR-378a-3p [93], miR-135b-3p [94], miR-1261 [95], miR-545-3p [91], and miR-489-3p [96] promoted ferroptosis by acting as suppressors of GPX4. Of these, the latter was found to be induced in gastric cancer cells by levobupivacaine. However, circIL4R knockdown was seen downregulating miR-541,3p, hence inhibiting ferroptosis, and the target gene in this case was also GPX4 [76].

Moreover, miR-387a-3p and miR-224-5p [97,98] enhanced ferroptosis via the suppression of SLC7A11, circSnx12, and FTH1 expression, respectively. The attenuator miRNAs of ferroptosis include miR-375 [99], miR-520-5p [100], and miR-128-3p [101]. Of these, the former three miRNAs (miR-387a-3p and miR-224-5p) work through SLC7A11 signaling, while the latter (miR-375, miR-520-5p, and miR-128-3p) work via the ACSL4 pathway and expression.

Overall, ferroptosis can be considered a novel approach for the treatment of cancer cells, in particular for apoptotic-resistant cells. The understanding of the role of miRNA in the modulation of this pathway could be crucial in identifying potential therapeutic interventions and enhancing treatment efficacy (Table 1).

Functional enrichment analysis of miRNA through the Kyoto Encyclopedia of Genes and Genomes (KEEG) highlighted the role of the target genes in ferroptosis, which is an important pathway in cancer development, followed by iron metabolism, lipid peroxidation, and antioxidant defense. According to the KEGG pathway analysis results, a significant portion of the common pathways were associated with ferroptosis pathways. The antioxidant defense pathway (as shown in red in Figure 2) was found to be the most significantly associated pathway. The 13 genes involved, starting from the 40 related miRNAs, are identified in the various databases according to the Venn diagram (Figure 2).

### 4.3. miRNA Therapeutics

In addition to the miRNAs described above, the exploiting of ferroptosis-linked miRNAs as a therapeutic opportunity is worth considering. Although a significant number of miRNAs with a pro- or anti-ferroptosis activity were described (Table 1), only a few of them have found a putative therapeutic application to date. Reported below the miRNAs that can be targeted for inducing ferroptosis are the drugs already used in clinical practice. Notably, these drugs are being used for clinical application other than cancer, but, more recently, their possible use in the treatment of various tumors is being considered.

Ketamine, an intravenous anaesthetic that exhibits anti-inflammatory activity and analgesic and antidepressant effects suppresses the viability and proliferation of liver cancer cells both in vitro and in vivo. It was shown to downregulate GPX4 expression and upregulate miR-214-3p levels through the lncPVT1/miR-214-3p/GPX4 regulatory axis, thus promoting ferroptosis in liver cancer cells [103]. Propofol, another common anaesthetic, was found to inhibit the expression of STAT3 and upregulate miR-125b-5p by acting on the miR-125b-5p/STAT3 axis. This activity induces ferroptosis in gastric cancer cells [106]. In addition, propofol markedly repressed the invasion and migration of gastric cancer cells. Both ketamine and propofol, however, raise some concerns about abuse and likely will not be approved for tumor treatments [108]. MiR-382-5p directly inhibits SLC7A11 expression, and this interaction is exploited and implemented by Lidocaine. This anaesthetic drug promotes ferroptosis by increasing miR-382-5p levels, which in turn causes a suppression of SLC7A11. The results obtained so far show a good potential for the use of Lidocaine in ovarian cancer and breast cancer cells [109]. However, Lidocaine has adverse reactions in the central nervous and cardiovascular systems, which needs to be considered before its application in clinical practice [108]. An analogue mechanism was observed in gastric cancer cells for levobupivacaine, which exerts its anticancer effect by upregulating miR-489-3p that, in turn, binds to the 3′-UTR of SLC7A11 and reduces its levels [96]. MiR-324-3p elicits interest in different kinds of tumors, thanks to its activity as a negative regulator of GPX4 by specifically binding to the 3′-UTR of GPX4. This is the case for breast cancer, which was treated with Metformin to target the mir-324-3p/GPX4 axis in a pro-ferroptotic manner [110]. In addition, Icariside II is a natural active flavonoid that affects this axis and promotes ferroptosis in renal cell carcinoma [111]. Curcumenol is another natural molecule able to act as a ferroptosis inducer via the lncRNA H19/miR-19b-3p/FTH1 axis. This compound acts on miR-19b-3p and causes lung cancer cell death both in vitro and in vivo [112].

The mechanisms just described may offer new ways to hit tumors. However, some limitations must be considered such as the low specificity on the target. Trying to overcome this problem, different groups combined miRNAs with nanoparticles, which allows the molecules to be conveyed in the desired sites [108]. Guo et al. combined miR-21-3p with gold nanoparticles and injected them into melanoma transplanted mice to target the ATF3/miR-21-3p/TXNRD1 axis. They not only observed an increase in ferroptosis (as predicted) but saw a limited immunogenicity and no therapeutic effects for the delivery system alone, thus indicating the great potential of this system [107]. Analogous tests were conducted by Luo et al. by combining miR-101-3p with nanocarriers and injecting them in lung cancer cells in mice. The results showed an increase in ROS levels and a decrease in GSH and GPX4, indicating that miR-101-3p was involved in ferroptosis. In addition, researchers noticed a better ability of the miR-101-3p plasmids transported by nanocarriers for engrafting into the cells [105].

## 5. Conclusions

The modulation of expression of miRNA is closely related to the degree of malignancy, drug resistance, and prognosis of tumors. Indeed, the expression of miRNAs can inhibit tumor cells through a variety of mechanisms, including inhibiting the cell cycle [67], inhibiting angiogenesis [112], promoting tumor immunity [52], promoting apoptosis [45], and ferroptosis [91]. Ferroptosis is a kind of programmed death caused by iron-dependent lipid peroxidation, which is different from apoptosis, necrosis, and other cell death modes. The discovery of new cell death modes has great significance for the treatment of tumor diseases. In light of the above, the combination ferroptosis–miRNAs could be a promising way forward to ameliorate cancer therapies, making them more specific and finely targetable. However, further studies must be conducted to understand the terms under which these systems can cause collateral effects and how to avoid them. 

## Figures and Tables

**Figure 1 jpm-13-00719-f001:**
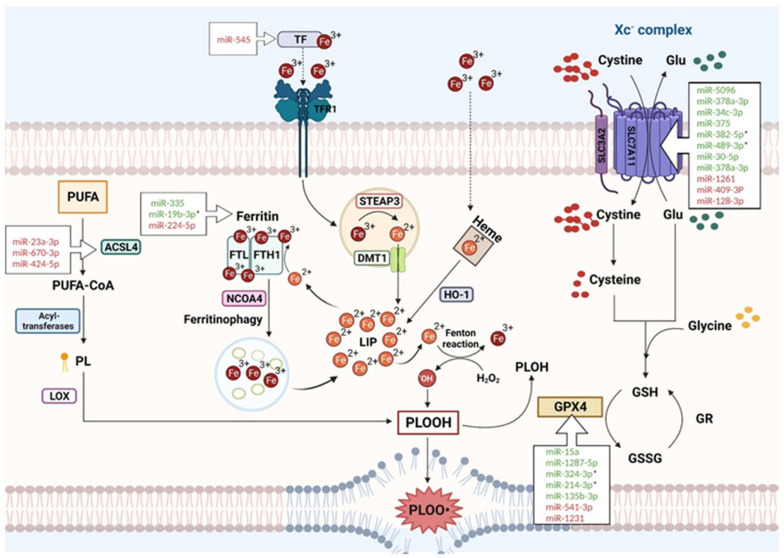
Interplay of the three main pathways contributing to ferroptosis and principal miRNAs targeting key ferroptotic proteins (see text for description). * miRNA reported as having a therapeutic impact. Created with https://www.BioRender.com (accessed on 20 April 2023).

**Figure 2 jpm-13-00719-f002:**
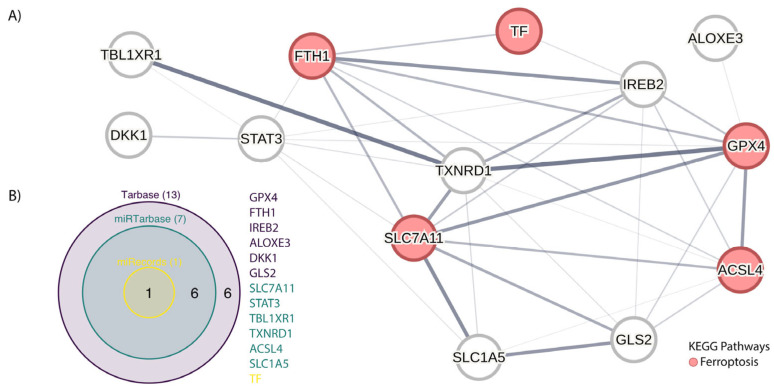
(**A**) Protein–protein interaction network for the main genes participating in the ferroptosis pathways, which represent the targets of the miRNAs reported in Table 1. The interactions show both functional and physical associations, so that one edge may represent the physical complex or another soft association. The analysis was conducted using STRING v11.5 software, setting the confidence parameter to 0.15 (edge thickness indicates the strength of data support: 0.15 thin and 0.9 thick). (**B**) Venn diagram for the validated targets in the Tarbased, miRTarbase, and miRecords database; the colors of the genes specify the database in which they are located. As shown, Tarbase collects all genes under study.

**Table 1 jpm-13-00719-t001:** Role of microRNA in the modulation of Ferropotosis.

MicroRNA	Gene	Type of Mechanism	Influence on Ferroptosis	References
miR-15a	GPX4	Egr-1/miR-15a-5p/GPX4 signaling pathway	Promotes ferroptosis	[67,68,70]
miR-1287-5p	GPX4	Phospholipid hydroperoxidase glutathione peroxidase GPX4 metabolism pathway	Promotes ferroptosis	[71]
miR-324–3p *	GPX4	Phospholipid hydroperoxidase glutathione peroxidase GPX4 metabolism pathway	Promotes ferroptosis	[72]
miR-214-3p *	GPX4	LncPVT1/miR-214-3p/GPX4 axis signaling pathway	Promotes ferroptosis	[102]
miR-182-5p	GPX4	NF-κB signaling pathway	Promotes ferroptosis	[93]
miR-135b-3p	GPX4	Inhibition of GPX4 expression	Promotes ferroptosis	[94]
miR-34c-3p	SLC7A11	Antioxidant defense	Promotes ferroptosis	[73]
miR-5096	SLC7A11	Targeting 3′UTR and downregulating SLC7A11	Promotes ferroptosis	[74]
MiR-378a-3p	SLC7A11	Antioxidant system	Promotes ferroptosis	[75]
miR-375	SLC7A11	Antioxidant system	Promotes ferroptosis	[76,99]
miR-143-3p	SLC7A11	Antioxidant defense	Promotes ferroptosis	[77]
miR-382-5p *	SLC7A11	Antioxidant system	Promotes ferroptosis	[103]
miR-489-3p *	SLC7A11	Targeting of SLC7A11 for downregulation	Promotes ferroptosis	[96]
miR-30-5p	SLC7A11	Antioxidant system	Promotes ferroptosis	[104]
miR-378a-3p	SLC7A11	Antioxidant system	Promotes ferroptosis	[93]
miR-101-3p *	TBLR1	NF-kB pathway	Promotes ferroptosis	[105]
miR-335	FTH1	Iron metabolism	Promotes ferroptosis	[78]
miR-19b-3p *	FTH1	Iron metabolism	Promotes ferroptosis	[106]
miR-122	IREB2	Iron metabolism	Promotes ferroptosis	[79]
miR-125b-5p *	STAT3	nd	Promotes ferroptosis	[100]
miR-21-3p *	TXNRD1	Lipid metabolism	Promotes ferroptosis	[107]
miR-19a	IREB2	Nrf2/HO-1 pathway	Inhibits ferroptosis	[87]
miR-23a-3p	ACSL4	Lipid metabolism	Inhibits ferroptosis	[80]
miR-670–3p	ACSL4	Lipid metabolism	Inhibits ferroptosis	[81]
miR-424–5p	ACSL4	Lipid metabolism	Inhibits ferroptosis	[83]
MiR-545	Transferrin (TF)	Iron metabolismLipid metabolism	Inhibits ferroptosis	[91]
miR-137	SLC1A5	Glutamine transporter	Inhibits ferroptosis	[86]
miR-130b-3p	DKK1	Nrf2/HO-1 pathway	Inhibits ferroptosis	[84,85]
miR-190a-5p	GLS2	Glutaminolysis pathway	Inhibits ferroptosis	[90]
miR-18	ALOXE3	YAP signaling pathway	Inhibits ferroptosis	[88,89]
miR-541-3p	GPX4	Antioxidant defense	Inhibits ferroptosis	[76]
miR-1231	GPX4	Antioxidant defense	Inhibits ferroptosis	[92]
miR-1261	SLC7A11	Circ0097009 regulates the expression of SLC7A11 by sponging miR-1261	Inhibits ferroptosis	[93]
miR-520-5p	SLC7A11	circFNDC3B increases SLC7A11 by targeting miR-520d-5p	Inhibits ferroptosis	[100]
miR-409-3p, miR-515-5p, miR-375	SLC7A11	Upregulation of SLC7A11 by circEPSTI1	Inhibits ferroptosis	[69]
miR-128-3p	SLC7A11	Antioxidant defense	Inhibits ferroptosis	[101]
miR-224-5p	FTH1	Iron metabolism	Inhibits ferroptosis	[97]

* miRNA reported as having a therapeutic impact.

## Data Availability

The datasets used and/or analyzed during the current study are available from the corresponding author on reasonable request.

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
