# Peer review of "Modulation of Ferroptosis by microRNAs in Human Cancer"

_jpm, 2023, doi:10.3390/jpm13050719_

Round 1
Reviewer 1 Report
Velkova and colleagues present an interesting review on the role of miRNAs in regulating cell death by ferroptosis. In general, I thoroughly enjoyed reading their review and found much of the information informative and accurate. There are only a few minor edits that I feel would improve the readability and utility of this work. They include:
1. The production of miRNAs and their modulation by DICER1 is not included and would really improve the introduction.
2. The whole introduction and section 2 and 2.1 are poorly referenced.
3. Section 2 focuses on oncogene driven mechanisms linked to ferroptosis. I would like this section expanded to include tumor suppressor genes, as a number of high-profile papers have recently linked p53 and pRB alterations to this process.
4. Please remove the word "seem" from this manuscript, this is a poor word choice.
5. Please add information of the toxicity of many of the therapies discussed in section 3.2
6. I feel that the statement "Previous studies have only linked......and cell death pathways (Line 246-247) is completely untrue, please remove.
7. Sections 4.1 and 4.2 read like a long list of facts, rather than a paragraph with independent ideas and findings. Please edit.
8. Please clearly outline the therapeutic strategies outlined in section 4.3 work via indirect mechanisms and may have additional roles on these processes.
9. Please include data on miRNA inhibitor studies.
Overall, I really liked this review and feel that with some work, it will be a real resource to the community.
The english in this manuscript is good, however it does require some minor changes.
Author Response
Dear Editors,
We would like to thank you for considering the manuscript entitled “Modulation of ferroptosis by microRNAs in human cancer” by Velkova I., Pasino M. et al. and for sharing the Reviewers’ comments that certainly helped in improving the quality of the manuscript (jpm-2355955). We appreciated the Reviewers’ comments, and we revised the manuscript accordingly. Please find enclosed to the submission of the revised version of the manuscript, the point-by point reply to the Reviewers’ comments. For clarity’s sake, changes in the revised MS are marked in yellow.
We hope that the revised version of our MS will be now suitable for publication in the JPM.
Reviewer 1:
COMMENT 1. The production of miRNAs and their modulation by DICER1 is not included and would really improve the introduction.
ANSWER 1. We are grateful with the Reviewer for this suggestion. We have improved the introduction as suggested.
COMMENT 2. The whole introduction and section 2 and 2.1 are poorly referenced.
ANSWER 2. We thank the Reviewer for the note. We have inserted appropriate references.
COMMENT 3. Section 2 focuses on oncogene driven mechanisms linked to ferroptosis. I would like this section expanded to include tumor suppressor genes, as a number of high-profile papers have recently linked p53 and pRB alterations to this process.
ANSWER 3. We thank the Reviewer for this suggestion. In Section 2 we added a short description of the involvement of p53 in the ferroptotic pathway. As we mentioned in the MS, the role of p53 in ferroptosis is rather complex implying both a promotion as well as a suppression of the process, likely in a cell-context dependent manner. More importantly, the role of p53 in ferroptosis involve a plethora of finely tuned proteins and pathways whose regulation is far from being the topic of this review. Certainly, it is a very interesting issue, but more specific and focused reviews are dedicated to this topic. Regarding pRB, it was not mentioned since a less clear role in ferroptosis has been suggested for the RB tumour suppressor which, together with E2F, represent a critical regulator of cell cycle progression. To our opinion, a discussion of its impact on ferroptosis would have taken us away from the main topic of the review itself.
COMMENT 4. Please remove the word "seem" from this manuscript, this is a poor word choice.
ANSWER 4. We thank the Reviewer for the note. The word has been removed from the whole manuscript or changed with appropriate synonyms, when necessary.
COMMENT 5. Please add information of the toxicity of many of the therapies discussed in section 3.2.
ANSWER 5. When available, the information regarding the toxicity of the therapies have been included.
COMMENT 6. I feel that the statement "Previous studies have only linked......and cell death pathways (Line 246-247) is completely untrue, please remove.
ANSWER 6. We thank the Reviewer for the note. The statement has been removed.
COMMENT 7. Sections 4.1 and 4.2 read like a long list of facts, rather than a paragraph with independent ideas and findings. Please edit.
ANSWER 7. We are grateful with the Reviewer for the suggestions.
COMMENT 8. Please clearly outline the therapeutic strategies outlined in section 4.3 work via indirect mechanisms and may have additional roles on these processes.
ANSWER 8. According to the reviewer suggestions, we added some details regarding additional roles that some therapeutic strategies may have. For two cited drugs, i.e. ketamine and propofol, we add other activities indirectly correlated with ferroptosis induction. Moreover, we also cited the concern about the application of some therapies in clinical practice due to their possible adverse effects. With these comments, we hope we have answered the reviewer's request correctly.
COMMENT 9. Please include data on miRNA inhibitor studies.
ANSWER 9. According to the reviewer suggestions, we added data on miRNA inhibitor studies.

Reviewer 2 Report
Comments to the authors:
This manuscript provides an overview of ferroptosis and related miRNAs in cancers. Overall, the topics presented in this manuscript are meaningful; however, some sentences and references need to be revised or reorganized. The suggestions are provided as follows:
1. In line 183-184, this paragraph describes cisplatin-induced apoptosis, which seems irrelevant to section 3.1 "Targeting the iron metabolism."
2. In section 4.2, "MicroRNAs in Ferroptosis Inhibition," some sentences describe how miRNAs promote ferroptosis. The paragraphs should be reorganized.
3. In line 337-339, the authors describe that the attenuator miRNAs of ferroptosis include miR-375 [90], miR-520-5p [91], and miR-128-3p [92]. Of these, the former three miRNAs work through SLC7A11 signalling, while the later via ACSL4 pathway/expression. It is unclear which miRNA is "the latter." Please clarify.
4. The article is entitled "Modulation of Ferroptosis by MicroRNAs in Human Cancer," yet some references describe the functions of miRNAs in other diseases, such as neurodegenerative diseases. It should be considered whether these studies are appropriate to discuss in this article.
5. The resolution of the figures should be improved.
6. The manuscript should be thoroughly checked for typos, such as “aminoacids”.
Author Response
Reviewer 2:
COMMENT 1. In line 183-184, this paragraph describes cisplatin-induced apoptosis, which seems irrelevant to section 3.1 "Targeting the iron metabolism."
ANSWER 1. We thank the Reviewer for the note. We have implemented it, underlying the role of the cisplatin and the promotion of the ferroptosis in lung cancer cells.
COMMENT 2. In section 4.2, "MicroRNAs in Ferroptosis Inhibition," some sentences describe how miRNAs promote ferroptosis. The paragraphs should be reorganized.
ANSWER 2. We are grateful with the Reviewer for the suggestions. The paragraphs have been reorganized (corrected the mistakes).
COMMENT 3. In line 337-339, the authors describe that the attenuator miRNAs of ferroptosis include miR-375 [90], miR-520-5p [91], and miR-128-3p [92]. Of these, the former three miRNAs work through SLC7A11 signalling, while the later via ACSL4 pathway/expression. It is unclear which miRNA is "the latter." Please clarify.
ANSWER 3. We are grateful with the Reviewer for the suggestions. The sentences are clarified by adding miRNAs in brackets.
COMMENT 4. The article is entitled "Modulation of Ferroptosis by MicroRNAs in Human Cancer," yet some references describe the functions of miRNAs in other diseases, such as neurodegenerative diseases. It should be considered whether these studies are appropriate to discuss in this article.
ANSWER 4. We thank the Reviewer for the note. We are aware that in two cases we reported studies not focused on cancer, but on disfunctions of the nervous system. However, we think that these references give interesting information regarding miRNAs that are able to modulate two main players in ferroptotic pathways, such as SLC7A11 and FTH1. Moreover, both miR-335 and miR378a-3p are implicated in cancer and considered as putative biomarkers in some circumstances. Thus, we decided not to remove them from the MS.
COMMENT 5. The resolution of the figures should be improved.
ANSWER 5. The resolution of the Figure has been improved as requested.
COMMENT 6. The manuscript should be thoroughly checked for typos, such as “aminoacids”.
ANSWER 6. We thank the Reviewer for the note. The manuscript has been revised and typos have been thoroughly checked.
